# Self-Supervised ImageNet Representations for In Vivo Confocal Microscopy: Tortuosity Grading without Segmentation Maps

**Kim Ouan**[1,2,3,4] (iD)                   KIM.OUAN@UNI-KOELN.DE
**Noémie Moreau**[1,2,3,4]                   NMOREAU@UNI-KOELN.DE
**Katarzyna Bozek**[1,2,3,4]                  K.BOZEK@UNI-KOELN.DE

[1] *Faculty of Mathematics and Natural Sciences, University of Cologne, Germany*

[2] *Institute for Biomedical Informatics, Faculty of Medicine and University Hospital Cologne, University of Cologne, Germany*

[3] *Center for Molecular Medicine Cologne (CMMC), Faculty of Medicine and University Hospital Cologne, University of Cologne, Germany*

[4] *Cologne Excellence Cluster on Cellular Stress Responses in Aging-Associated Diseases (CECAD), University of Cologne, Germany*

## Abstract

The tortuosity of corneal nerve fibers are used as indication for different diseases. Current state-of-the-art methods for grading the tortuosity heavily rely on expensive segmentation maps of these nerve fibers. In this paper, we demonstrate that self-supervised pretrained features from ImageNet are transferable to the domain of in vivo confocal microscopy. We show that DINO should not be disregarded as a deep learning model for medical imaging, although it was superseded by two later versions. After careful fine-tuning, DINO improves upon the state-of-the-art in terms of accuracy (84,25%) and sensitivity (77,97%). Our fine-tuned model focuses on the key morphological elements in grading without the use of segmentation maps. The code is available at https://github.com/bozeklab/tortuosity-dino.
**Keywords:** In Vivo Confocal Microscopy, Corneal Nerve Fibers, Tortuosity, Self-supervised Learning

## 1. Introduction

The tortuosity of corneal nerve fibers serves as an indication for several diseases (Lagali et al., 2015). In recent years, deep learning models for classifying the tortuosity have achieved better results than classical approaches (Mou et al., 2022; Colonna and Scarpa, 2023). Yet, these methods, do not only require tortuosity labels but also segmentation maps (Zhao et al., 2020; Colonna and Scarpa, 2023; Mou et al., 2022) which are laborious to generate.

In this study, we show that the representations from a self-supervised ImageNet pre-trained DINO (Russakovsky et al., 2015; Caron et al., 2021) exhibit strong classification performance on the tortuosity classification task of in vivo confocal microscopy (IVCM) images using linear probing, albeit the different domain. In this task, it beats its successors, DINOv2 (Oquab et al., 2024) and DINOv3 (Siméoni et al., 2025). Moreover, our fine-tuned DINO improves upon the previous state-of-the-art in terms of accuracy and sensitivity, with

Figure 1: (a) Example images of the CORN$^{1500}$ dataset. Ordered from the mildest level 1 (left) to the most severe level 4 (right). (b) UMAP visualization of CORN$^{1500}$ features extracted by the frozen DINO backbone • level 1, • level 2, • level 3, • level 4.

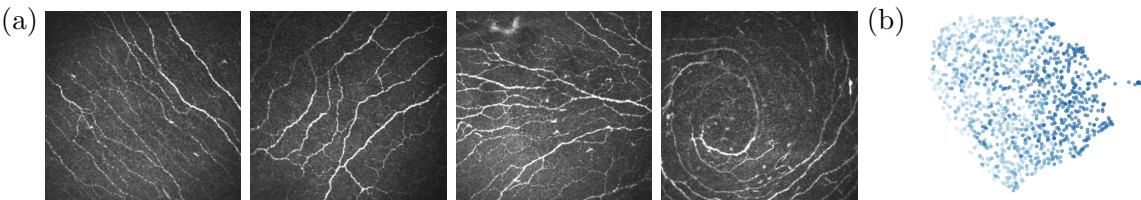

a trade-off in specificity under the same evaluation protocol. In contrast to previous approaches, our method does not require additional segmentation maps for the classification of tortuosity.

## 2. Data and Experiments

We use the data sets CORN$^{1500}$ (Mou et al., 2022) and CORN-3 (Zhao et al., 2020). Both data sets divide the severity of tortuosity into a discrete scale of four levels. The images are of size $384 \times 384$ pixels and cover an area of $400 \times 400 \ \mu m^2$. CORN$^{1500}$ contains 1500 images distributed across the four levels: 214, 461, 364, and 461, respectively. CORN-3 contains 403 images distributed across the four levels: 54, 212, 108, and 29, respectively (Mou et al., 2022). In Figure 1 (a) example images for each level are visualized.

In our experimental setup, we use the same data split – CORN$^{1500}$ for training and validation, CORN-3 for testing – and report the same metrics as Mou et al. (2022). In linear probing, we attach one linear layer on top of the frozen backbone and optimize its parameters for 100 epochs using stochastic gradient descent with an initial learning rate of 0.01, a cosine decay learning rate schedule and a momentum of 0.9. We compare linear probing results across different DINO versions (DINO/DINOv2/DINOv3) and use the best performing model - DINO ViT-B - for fine-tuning. During fine-tuning, we freeze the bottom half of the backbone for 30 epochs, attach one linear layer on top of the backbone and use AdamW (Loshchilov and Hutter, 2019) optimizer with a linear warm-up scheduler to the peak learning rate of 0.0001, followed by a cosine decay learning rate schedule and a layerwise decay, following the pre-training implementation of DINOv2 (Oquab et al., 2024), and apply early stopping. Fine-tuning the model on one NVIDIA Tesla V100-SXM2-32GB GPU for a total of 200 epochs takes approximately 90 minutes.

## 3. Results and Conclusion

Table 1 (a) shows that DINO performs better than it's successors DINOv2 and DINOv3 in linear probing on IVCM data. The embeddings extracted from the ImageNet pre-trained frozen DINO backbone are visualized in Figure 1 (b). Without the model ever having seen IVCM data, the embeddings arrange along a gradient from left to right, compatible with their assigned tortuosity level. Figure 2 (a) shows that the attention maps of the frozen backbone highlight the corneal nerve fibers despite being trained on natural images

Table 1: **Evaluation on CORN-3**. All metrics – weighted accuracy (wAcc), weighted sensitivity (wSe), weighted specificity (wSpe) – are shown in percentage (%). The highest metrics are highlighted in **bold**. For the per level comparison and the comparison of different model sizes see Table 2 and Table 3 in Appendix A. (a) Linear probing results of DINO/v2/v3 ViT-B. (b) Comparison with state-of-the-art methods. Our fine-tuned DINO achieves state-of-the-art results without the need of segmentation maps (SM).

| Model | wAcc | wSe | wSp |
|---|---|---|---|
| DINO | **78.63** | **69.55** | 81.51 |
| DINOv2 | 67.53 | 52.23 | 78.49 |
| DINOv3 | 74.28 | 61.88 | **84.15** |

(a)

| Model | SM | wAcc | wSe | wSp |
|---|---|---|---|---|
| M4 (Zhao et al., 2020) | ✓ | 82.40 | 71.70 | 85.90 |
| DeepGrading (Mou et al., 2022) | ✓ | 84.10 | 76.18 | **90.71** |
| DINO ViT-B/16 (fine-tuned) | ✗ | **84.25** | **77.97** | 84.81 |

(b)

Figure 2: (a) Attention masks over CORN-3 IVCM images of level 1 (left) and level 4 (right) for 60% of the attention mass of the last layer of the encoder. (b) The confusion matrix of our fine-tuned model shows only misclassifications in adjacent levels.

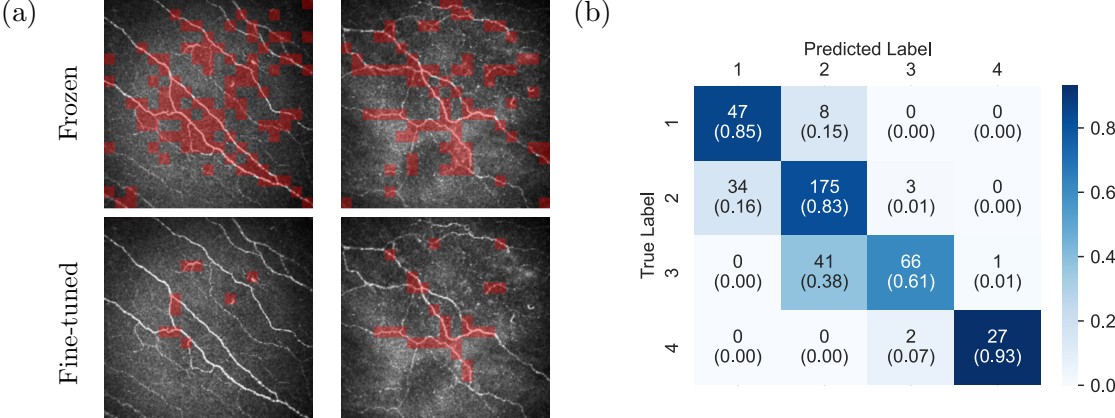

only. After fine-tuning, the reported metrics in Table 1 (b) show that our model achieves state-of-the-art performance. Concurrently, the fine-tuned model's attention is focused on small parts of the nerve fibers that indicate the tortuosity – such as branching points and segments with high curvatures. Additionally, the confusion matrix of our final model in Figure 2 (b) indicates that the model captures the underlying ordinal structure of the data as the misclassifications are within adjacent classes only.

Our findings show that DINO solely pre-trained on ImageNet, creates strong representations for tortuosity grading in IVCM images. We demonstrated that the fine-tuned DINO reaches state-of-the-art performance for tortuosity grading and focuses on the key parts of the image without the need of segmentation maps.

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

# Appendix A. Detailed Evaluation on CORN-3

Table 2: Comparison of linear probing for different model sizes of DINO, DINOv2, and DINOv3. Both DINO models show the best balance between the different metrics and levels. All metrics are shown in percentage (%).

| Model | Size | Metrics | *level1* | *level2* | *level3* | *level4* | **overall** |
|---|---|---|---|---|---|---|---|
| DINO | ViT-S/16 | *wAcc* | **88.86** | **72.28** | 79.70 | 96.78 | 78.28 |
| | | *wSe* | 87.27 | **80.19** | 37.96 | 65.52 | 68.81 |
| | | *wSp* | **89.11** | 63.54 | 94.93 | 99.20 | 77.97 |
| | ViT-B/16 | *wAcc* | 87.62 | 72.03 | 81.93 | 97.52 | **78.63** |
| | | *wSe* | 89.09 | 72.64 | 49.07 | **86.21** | **69.55** |
| | | *wSp* | 87.39 | 71.35 | 93.92 | 98.40 | 81.51 |
| DINOv2 | ViT-S/14 | *wAcc* | 80.05 | 63.12 | 77.72 | 97.03 | 71.82 |
| | | *wSe* | 92.73 | 64.62 | 24.07 | **86.21** | 59.16 |
| | | *wSp* | 78.51 | 61.46 | **97.30** | 97.87 | 75.97 |
| | ViT-B/14 | *wAcc* | 70.54 | 56.19 | 80.20 | 97.52 | 67.53 |
| | | *wSe* | **100.00** | 43.87 | 35.19 | 86.21 | 52.23 |
| | | *wSp* | 65.90 | 69.79 | 96.62 | 98.40 | 78.49 |
| | ViT-L/14 | *wAcc* | 82.92 | 69.31 | 82.67 | 97.77 | 76.78 |
| | | *wSe* | 96.36 | 55.19 | **70.37** | 75.86 | 66.34 |
| | | *wSp* | 80.80 | **84.90** | 87.16 | **99.47** | **85.99** |
| DINOv3 | ViT-S/16 | *wAcc* | 81.19 | 66.01 | 81.93 | 97.52 | 74.64 |
| | | *wSe* | 94.55 | 59.91 | 50.93 | 75.86 | 63.37 |
| | | *wSp* | 79.08 | 72.92 | 93.24 | 99.20 | 81.08 |
| | ViT-B/16 | *wAcc* | 76.24 | 65.84 | **83.42** | **98.27** | 74.28 |
| | | *wSe* | 98.18 | 52.83 | 55.56 | 82.76 | 61.88 |
| | | *wSp* | 72.78 | 80.21 | 93.58 | **99.47** | 84.15 |
| | ViT-L/16 | *wAcc* | 80.69 | 65.84 | 79.70 | 96.04 | 73.74 |
| | | *wSe* | 96.36 | 60.38 | 41.67 | 72.41 | 61.14 |
| | | *wSp* | 78.22 | 71.88 | 93.58 | 97.87 | 80.41 |

The results for the evaluation of different DINO models and model sizes using linear probing for tortuosity level classification of IVCM images are reported in Table 2. It shows that DINO ViT-B/16 exhibits the best balance between all metrics and levels. Table 3 reports the full metric table of our fine-tuned DINO compared with state-of-the-art methods for tortuosity classification. It is shown the fine-tuned DINO achieves state-of-the-art performance without using additional segmentation maps.

Table 3: **Evaluation on CORN-3**. Comparison with state-of-the-art methods. Our fine-tuned DINO achieves state-of-the-art results without the need of segmentation maps. The metrics are shown in percentage (%). The highest metrics are highlighted in **bold**.

| Methods | SM | Metrics | *level1* | *level2* | *level3* | *level4* | **overall** |
|---|---|---|---|---|---|---|---|
| Annunziata (Annunziata et al., 2016) | ✓ | *wAcc* | 85.80 | 78.00 | 75.90 | 84.30 | 79.00 |
| | | *wSe* | 71.80 | 64.40 | 66.00 | 70.70 | 66.30 |
| | | *wSp* | 86.70 | 78.00 | 75.90 | 84.30 | 79.00 |
| M4 (Zhao et al., 2020) | ✓ | *wAcc* | 88.40 | **80.30** | 80.00 | 86.60 | 82.40 |
| | | *wSe* | 74.30 | 68.00 | 68.80 | 73.80 | 71.70 |
| | | *wSp* | 90.10 | 80.10 | 81.40 | 88.10 | 85.90 |
| DeepGrading (Mou et al., 2022) | ✓ | *wAcc* | 86.85 | 79.90 | 87.10 | 98.51 | 84.10 |
| | | *wSe* | **98.15** | 70.75 | **72.22** | 89.66 | 76.18 |
| | | *wSp* | 85.10 | **90.05** | 92.54 | 99.20 | **90.71** |
| DINO ViT-B/16 (fine-tuned) | ✗ | *wAcc* | **89.60** | 78.71 | **88.37** | **99.26** | **84.25** |
| | | *wSe* | 85.45 | **82.55** | 61.11 | **93.10** | **77.97** |
| | | *wSp* | **90.26** | 74.48 | **98.31** | **99.73** | 84.81 |

Table 4: Evaluation metrics of our fine-tuned model on **CORN-3-noD**. The metrics are shown in percentage (%).

| Methods | Metrics | *level1* | *level2* | *level3* | *level4* | **overall** |
|---|---|---|---|---|---|---|
| DINO ViT-B/16 (fine-tuned) | *wAcc* | 86.93 | 76.38 | 88.94 | 99.50 | 83.71 |
| | *wSe* | 85.71 | 73.63 | 69.23 | 100.00 | 75.88 |
| | *wSp* | 87.13 | 78.70 | 98.51 | 99.46 | 87.92 |

## Appendix B. Reduced Data Sets

The results in Table 3 were obtained under the same experimental setup as Mou et al. (2022) to ensure comparability. However, we also removed artifacts and label ambiguities from this data set creating its new version that we named CORN$^{1500}$-noD and CORN-3-noD. CORN$^{1500}$-noD contains 1250 images distributed across the four levels: 188, 396, 289, 377, respectively. CORN-3-noD contains 199 images distributed across the four levels: 28, 91, 65, 15, respectively. We retrained our model on CORN$^{1500}$-noD with a data split of 70/30 for training and validation and evaluated it on CORN-3-noD. The final results on CORN-3-noD are shown in Table 4. The indices of all data samples used in CORN$^{1500}$-noD and CORN-3-noD are published in https://github.com/bozeklab/tortuosity-dino.

