# OpenReview forum: "Self-Supervised ImageNet Representations for In Vivo Confocal Microscopy: Tortuosity Grading without Segmentation Maps"
_MIDL.io/2026/Short_Papers — MIDL 2026 - Short Papers Poster_

### Official Review · Reviewer_FDfs · 2026-04-28
**good short paper**

**Rating:** 3
**Confidence:** 5

**Review:**

The paper addresses a relevant problem in medical imaging: reducing reliance on segmentation maps for downstream classification tasks. The idea of leveraging self-supervised pretrained representations is well motivated, and the experimental design—comparing linear probing, fine-tuning, and different DINO variants—is clear and easy to follow. The qualitative analysis (e.g., attention maps) further supports the claim that the model captures meaningful anatomical structures.

In terms of originality, novelty is very limited. The use of pretrained vision transformers such as DINO in medical imaging has been explored, and the main contribution here is demonstrating that an earlier version (DINO) can outperform its successors in this specific domain and task. However, investigating Table2 later models of DINO perform better in different levels also the best results shown in Table 2 do not match with the best results shown in Table 1.

**Summary:**

This work investigates the use of self-supervised DINO features pretrained on ImageNet for grading corneal nerve tortuosity in in vivo confocal microscopy images, without requiring segmentation maps. The authors demonstrate that both linear probing and fine-tuning of DINO achieve strong performance, surpassing more recent variants such as DINOv2 and DINOv3 in this task.

**Strengths:**

Demonstrates strong transferability of self-supervised features from natural images to medical imaging. Achieves state-of-the-art performance on tortuosity grading benchmarks. Clear and well-structured experimental design, including comparison across DINO variants.

**Weaknesses:**

Limited methodological novelty: the approach primarily relies on standard fine-tuning of DINO without introducing new architectural or learning innovations. Dataset size and diversity are limited (CORN1500 and CORN-3), raising concerns about robustness and generalization to other populations or imaging conditions. Comparison is restricted to a narrow set of baselines; more recent medical imaging-specific or foundation models (e.g., MedSAM-like approaches) are not evaluated. The claim that DINO outperforms newer variants (DINOv2, DINOv3) is interesting but not deeply analyzed—no investigation into why this occurs. Also results for best performing DINO models do not match between tables 1-2 Performance trade-offs are not fully explored: improved sensitivity comes at reduced specificity, which may be critical in clinical deployment.

**Justification Of Rating:**

PLease see above

---

### Decision · Program_Chairs · 2026-05-08

Accept (Poster)